# A 28-Day Repeated Oral Administration Study of Mechanically Fibrillated Cellulose Nanofibers According to OECD TG407

**DOI:** 10.3390/nano14131082

**Published:** 2024-06-24

**Authors:** Yoshihiro Yamashita, Akinori Tokunaga, Koji Aoki, Tamotsu Ishizuka, Satoshi Fujita, Shuichi Tanoue

**Affiliations:** 1Research Center for Fibers and Materials, University of Fukui, 3-9-1 Bunkyo, Fukui 910-8507, Japan; 2Life Science Research Laboratory, School of Medical Sciences, University of Fukui, 23-3, Matsuokashimoaizuki, Eiheiji-cho 910-1193, Japan; 3Department of Pharmacology, Faculty of Medicine, University of Fukui, 23-3, Matsuokashimoaizuki, Eiheiji-cho 910-1193, Japan; 4Department of Respiratory Medicine, Faculty of Medical Sciences, University of Fukui, 23-3, Matsuokashimoaizuki, Eiheiji-cho 910-1193, Japan; 5Department of Frontier Fiber Technology and Science, Faculty of Engineering, University of Fukui, 3-9-1 Bunkyo, Fukui 910-8507, Japan

**Keywords:** in vivo, OECD TG407, fibrillated cellulose nanofibers

## Abstract

The impact of oral administration of mechanically fibrillated cellulose nanofibers (fib-CNF), a commonly used nanofiber, on toxicity and health remains unclear, despite reports of the safety and beneficial effects of chitin-based nanofibers. Thus, evaluating the oral toxicity of fib-CNF in accordance with OECD Test Guideline 407 (TG407) is essential. This study aimed to assess the safety of orally administered fib-CNF through an acute toxicity study in rats, following the OECD TG407 guidelines for 4 weeks. CNF “BiNFi-s” FMa-10005, derived from mechanically fibrillated pulp cellulose, was administered via gavage to male and female Crl:CD(SD) rats at doses of 50, 150, 500, and 1000 mg/kg/day for 28 days, with a control group receiving water for injection. The study evaluated the toxic effects of repeated administration, and the rats were monitored for an additional 14 days post-administration to assess recovery from any toxic effects. The results showed no mortality in either sex during the administration period, and no toxicological effects related to the test substance were observed in various assessments, including general condition and behavioral function observations, urinalysis, hematological examination, blood biochemical examination, necropsy findings, organ weights, and histopathological examination. Notably, only female rats treated with 1000 mg/kg/day of CNF exhibited a consistent reduction in body weight during the 14-day recovery period after the end of treatment. They also showed a slight decrease in pituitary and liver weights. However, hematological and blood biochemical tests did not reveal significant differences, suggesting a potential weight-suppressive effect of CNF ingestion.

## 1. Introduction

The safety evaluation of nanomaterials is crucial for both workers and consumers [1]. Recently, cellulose nanofibers (CNFs) have attracted attention not only as reinforcements for composite materials but also as potential food additives. The safety of cellulose nanomaterials in food applications has been researched extensively [2,3,4,5,6]. Chen et al. [7] noted that CNFs nonspecifically decrease intestinal absorption, potentially posing a nutritional risk after long-term exposure. Carboxymethylated CNF (CMC) has been widely used as a food additive [8]. Zhang et al. [9] administered 1% or 3.5% w/w% of CMC to mice via gavage once daily for 8 weeks. They found no significant differences in hematological and serum markers between CMC-fed mice and control mice. However, CMC-fed mice exhibited a significant increase in fecal fat compared to controls, along with decreased food intake and body weight. These findings suggest that CNF is non-toxic and may function as a food additive or supplement to reduce caloric intake.

One method for producing nanofiber-sized CNFs involves repeatedly impinging pulp with high-speed fluid, a technique known as the starburst method. This approach produces mechanically fibrillated cellulose nanofibers (fib-CNF). Xu et al. [10] reported that administering CNF to NOD mice at a dose of 30 mg/kg for 6 months resulted in a significant increase in immobility time during the suspension test, suggesting increased depression-related behavior.

Several animal studies have investigated the effects of chitin nanofibers (CS) with CNFs. Azuma et al. [11] fed rats a high-cholesterol diet containing 1% w/w cholesterol and 0.5% w/w cholic acid for 28 days. During the experimental period, the control group received 0.05% acetic acid diluted in tap water, while groups receiving deacetylated chitin nanofibers (SDACNF), chitosan (CS), and CNFs had their solutions adjusted to 0.1% in drinking water, which they consumed ad libitum. Administration of SDACNF and CS reduced diet-induced increases in serum levels of total cholesterol, chylomicrons, very low-density lipoprotein, and phospholipids on day 14. Furthermore, oral administration of SDACNF inhibited the elevation of alanine transaminase levels by day 29 and prevented the accumulation of vacuolar degeneration and fat droplets in liver tissue. These findings suggest that SDACNF, similar to CS, may lower blood cholesterol levels through adsorption. This indicates that the reduction in blood lipid levels was not solely due to the lipid adsorption effect of CS but also due to its influence on lipid metabolism. The study suggests that modified chitosan nanofibers may enhance biological health.

On the other hand, Shimokawa et al. [12] prepared CNFs from bamboo pulp produced by soda evaporation using an enzymatic wet cracking method, aiming to apply Phyllostachys edulis to food materials. They found that this bamboo-derived CNF was more nano-sized than microfibrous cellulose, which is recognized as a food additive in Japan, and did not fit into the microfibrous cellulose category. Consequently, a 90-day repeated oral toxicity study was conducted on rats using a 1 wt% suspension of bamboo CNF, and no deaths were observed in any of the groups during the 90-day administration period. Hematology, blood biochemistry, organ weights, necropsy, and histopathology performed at the end of the treatment period showed no effects from the test substance administration. Since no toxicity was observed in either male or female animals, the no-observed-adverse-effect Level (NOAEL) of CNF from bamboo was determined to be 200 mg/kg/day or higher for both sexes. However, this experiment did not adhere to OECD guidelines.

Given the potential use of fib-CNF as a food additive, we conducted an oral acute toxicity study (TG407) in accordance with OECD guidelines to assess its safety. The impact of oral administration of fib-CNF, widely used among nanofibers, remains unclear, with reports suggesting both toxicity and health benefits, similar to the documented safety and beneficial effects of chitin-based nanofibers [13]. Thus, adherence to the guidelines outlined in OECD TG407 for evaluating oral toxicity is essential [14,15]. Furthermore, no studies utilizing TG407 for orally administered nanofibers have been reported, and research on the safety of orally administered nanofibers has typically been limited to 2 weeks or low concentrations [16,17]. Consequently, this study aimed to assess the safety of orally administered fib-CNF through a four-week acute toxicity study in rats, in accordance with OECD TG407 guidelines.

## 2. Experiments

### 2.1. Fib-CNF

Fib-CNFs extracted from wood cellulose pulp were used in this study following sterilization at 80 °C (fib-CNF, BiNfi-s FMa-10005, Lot D2023601, Sugino Machine, Toyama, Japan). The potential impurities in fib-CNF (BiNFi-s) were as follows: general bacteria <10 colony-forming units/mg, *E. coli* negative, ash < 0.1 wt% (Hygienic Testing Method Commentary 2015, Tokyo, Japan), heavy metals < 0.01 mg/kg (ICP-MS), and arsenic < 0.01 mg/kg (ICP-MS). To improve the stability of the fib-CNF aqueous solution, a dispersion solution that did not sediment even after standing for 1 week was obtained via high-speed stirring at 8000 rpm for 20 min using a homogenizer (HG-200; AS ONE Corporation, Osaka, Japan). The fiber length and width of fib-CNF were measured using a Hitachi H-7650 transmission electron microscope (acceleration voltage 100 kV) and a JEOL (Tokyo, Japan) JSM-7600F field emission scanning electron microscope (acceleration voltage 10 kV), respectively, along with ImageJ (ImageJ-win64) analysis software. This CNF was prepared as a 5% pure water solution of “BiNFi-s” FMa-10005, which was diluted as necessary with pure Japanese Pharmacopoeia water for injection (hereinafter referred to as the “medium”; Otsuka Pharmaceutical Factory Co., Tokyo, Japan, 2D74N, 2E92N).

### 2.2. Rats

Forty-four 5-week-old male and female Crl:CD(SD) rats were purchased from Jackson Laboratory Japan, Inc. (Atsugi, Japan). They were housed under specific pathogen-free (SPF) conditions, with forty rats selected for the experiment. The rats underwent an 8-day quarantine and acclimation period during which they were weighed twice and then divided into groups. They were fed ad libitum with solid diets MF (220408, 220514, 220609 by Oriental Yeast Co., Tokyo, Japan). The experimental design, including gender, dose of the test substance, dosing volume, dose concentration, number of animals per group, and animal numbers, is shown in Table 1. Based on OECD TG407, the limit dose was set at 1000 mg/kg body weight/day. The high dose was 500 mg/kg/day, and the low doses were set at 150 mg/kg/day and 50 mg/kg/day, with a common ratio of approximately 3 between doses. The group receiving 1000 mg/kg/day was administered the undiluted 5 wt% CNF solution. The other groups received the test substance diluted with pure water as described in Section 2.1. The maximum volume that can be administered to a rat’s stomach at one time is 10 mL; at 5 wt% CNF, this equates to a CNF dose of 500 mg in solid form. Therefore, the dose was divided into two administrations. The medium and low concentrations were also divided into two doses to ensure proper administration.

### 2.3. Administration Method

The OECD Guideline TG407 specifies that the test substance should be administered orally to experimental animals at graded doses (one dose per group) over a period of 28 days. These 28-day studies provide information on the effects of repeated oral exposure. The TG407 study allows for the characterization of the test substance’s toxicity, the clarification of a dose–response relationship, and the determination of a no-observed-adverse-effect level (NOAEL). Furthermore, TG407 recommends using at least 10 animals (5 females and 5 males) for each dose group, as well as adding a satellite group of 10 animals (5 females and 5 males) in both the control and highest dose groups to observe for at least 14 days after administration completion. This is to determine the reversibility, persistence, or delayed onset of toxic effects. The numbers of dosing and recovery days were established according to these guidelines.

The oral route was chosen because humans typically ingest the test substance orally. Forced oral administration ensured the precise delivery of the prescribed dose. The maximum dose group received the undiluted solution (5% fib-CNF). The diluted solution was administered to the high-dose group at a concentration of 2.5% CNF, the medium-dose group at 0.75%, and the low-dose group at 0.25%. For dilution, the required volume of the test substance was measured with a graduated cylinder, and pure water was added to reach the desired volume. Since the stability of the dosing solution had not been confirmed, the solution was prepared immediately before administration. Although the CNF concentration in the dosing solution was not verified, the test substance preparation record ensured correct solution preparation.

After grouping the animals, the administration of the test substance commenced at 6 weeks of age, starting the day before the experiment began. The test substance was administered twice daily (7 days per week) for 28 days, with the second dose given 3 to 4 h after the first dose. Doses were administered by forced oral gavage at a volume of 10 mL/kg using a disposable syringe and transgastric tube. Doses were calculated individually based on body weight. A 14-day recovery period was allowed for the medium control, high-dose, and limit-dose groups after dosing completion.

### 2.4. Health Observations

After the experiment began, all animals were carefully observed twice daily (after each dose during the dosing period, and in the morning and afternoon during the recovery period) for their general condition, toxic symptoms, and vital status, with observations recorded for each individual animal.

### 2.5. Body Weight

All animals were weighed using an LA4200 electronic balance (Sartorius Corporation, Tokyo, Japan) at the beginning of administration, once a week thereafter during the administration period, on the first day of recovery, and once a week during the recovery period. The weight of each animal after an overnight fast (approximately 17 h) was also determined at the time of planned necropsy (necropsy weight).

### 2.6. Food Intake

The average daily food intake per animal was calculated by measuring the amount of food consumed once a week for 2 days using an LA4200 electronic balance (Sartorius Co., Ltd., Göttingen, Germany) on a per-cage basis.

### 2.7. Urinalysis

The tests listed in Table 2 were conducted on all animals after 4 weeks of treatment (including the recovery group) and again after 2 weeks of recovery. On the day of testing, following administration, animals were transferred to a urine collection rack (Jackson Laboratory Japan, Inc.) to begin urine collection. Feed and drinking water were provided using a feeder and water bottle specifically designed for the urine collection rack. Transfer to the urine collection rack was performed in groups after administration. Approximately 4 h (+1 to +10 min) later, a urine sample was collected, and the urine collection container was replaced. This was followed by a storage period of approximately 20 h (−6 to −5 min) before further analysis. Items 1 to 8 were assessed using freshly collected urine obtained after 4 h of storage, and items 9 to 11 were analyzed using urine stored for approximately 20 h. The measurement items and methods are detailed in Table 2 below.

### 2.8. Hematological Examinations

The tests detailed in Table 3 were conducted on all animals following the conclusion of the test substance administration period (excluding the recovery group) and after the recovery period. Animals were fasted from the evening before the collection (around 16:00). On the day of collection, anesthesia with isoflurane (Mylan Seiyaku, Tokyo, Japan, Lot No. 237KKS) was administered. The abdomen was opened using NARCOBIT-E (KN-1070, Natsume Manufacturing Co., Ltd., Tokyo, Japan), a simple inhalation anesthetic device for small laboratory animals. Blood samples were collected from the large abdominal artery after the removal of the abdomen. The animals were euthanized via exsanguination from the abdominal aorta.

The order of blood collection (euthanasia) proceeded as follows: the youngest animal in the first group was processed first, followed by the youngest animal in the next group, and so on, rotating back to the first group after the last group was reached. A portion of the collected blood was transferred to a test tube containing an anticoagulant. Items 1 to 10 were measured using an XT-2000i multi-parameter automated hematology analyzer (Sysmex Corporation, Kobe, Japan), and items 11 and 12 were measured using a CA-530 fully automated coagulation analyzer (Sysmex Corporation). For anticoagulants, EDTA-2K was used for items 1 to 10, and sodium citrate was used for items 11 and 12.

### 2.9. Blood Biochemical Examinations

Serum, obtained through the centrifugation of the remaining blood collected in the previous section, was utilized for testing the items listed in Table 4. The measuring instrument employed was the Hitachi Automatic Analyzer 3500 (Hitachi High-Technologies Corporation, Tokyo, Japan).

### 2.10. Pathological Examination

At the end of the dosing period or recovery period, blood was collected, and the animals were euthanized for a thorough pathological examination, as described in the following section.

### 2.11. Necropsy

All animals underwent a comprehensive gross pathological examination of various organs and tissues. The following organs and tissues were removed and fixed in a 10% buffered formalin solution, except for the testes (pre-fixed in Glutaraldehyde Formalin Aceticid (GFA) solution) and the eyeballs (fixed in Glutaraldehyde-Formaldehyde (G-FFA) solution, and a formaldehyde (G-F) mixture). The organs included were the heart, spleen, lymph nodes (cervical and mesenteric), thymus, pituitary gland, thyroid gland, adrenal gland, trachea, lung (including bronchi), salivary glands (submandibular and sublingual glands), esophagus, stomach, small intestine, large intestine, liver, pancreas, kidney, urinary bladder, testis, epididymis, prostate, seminal vesicle, ovary (including fallopian tubes), uterus, mammary gland, vagina, brain (cerebrum and cerebellum), spinal cord (cervical, thoracic, and lumbar), sciatic nerve, aorta, eyes, skin, bone and marrow (sternum and femur), skeletal muscle, and other gross lesions.

### 2.12. Histopathological Examination

For the control and limit-dose groups (excluding the recovery group), specimens were prepared by embedding them in paraffin. The tissue sections were stained with hematoxylin and eosin and analyzed under a microscope. In the other groups, only gross lesions were examined.

### 2.13. Statistical Treatment

Significance tests were conducted between the control group and each treatment group at a significance level of 5% (*p* < 0.05) or 1% (*p* < 0.01). Measurements included behavioral function observations (grip and locomotion), body weight, food intake, urinalysis (excluding urine test strips, urine sediment, and color), hematology, blood biochemistry, and organ weights. Equal variance tests were performed using the Bartlett’s test at the 5% significance level during the treatment period. In the case of equal variances, the parametric Dunnett’s test (two-tailed) was employed. In the case of unequal variances, the nonparametric Steel’s test (two-tailed) was used. During the recovery period, the F-test was applied, and in cases of equal variance, the Student’s *t*-test (two-tailed) was performed. In instances of unequal variance, the Welch’s test (two-tailed) was conducted. Fisher’s exact test (one-tailed) was used to assess differences in the frequency of occurrences during necropsy and histopathological examination. For tests related to differences in urinalysis (urine dipstick test, urine sediment, and color), a chi-square test (two-tailed) was performed on both males and females. If significant differences were identified between groups, a chi-square test (two-tailed) was also performed on the control group and each dose group. No statistical analysis was applied to general condition and behavioral function observations, except for grip strength and locomotor activity.

## 3. Results

### 3.1. Fiber Length and Fiber Diameter of Fib-CNF

The measurement outcomes are shown in Figure 1. The mean fiber length was 756 nm (Standard Deviation [S.D.] 306 nm, *R*^2^: 0.45; Figure 1a), while the average fiber diameter was 42.9 nm (S.D. 9.2 nm, *R*^2^: 0.46 Figure 1b). The average fiber diameter closely aligned with the manufacturer’s catalog value. It is noteworthy that fib-CNFs exhibit a morphology distinct from needle-like structures; instead, they assume the form of short buckwheat noodles (Figure 1c).

### 3.2. Weight

The OECD Guideline TG407 specifies that the test substance is administered orally to experimental animals at graded doses (one dose per group) over a period of 28 days. Furthermore, the TG407 guideline recommends observing the highest dose groups for at least 14 days after the completion of administration to analyze the reversibility, persistence, or delayed onset of toxic effects. The number of dosing and recovery days was determined according to this guideline. The mean body weight and standard deviation for each group are shown in Figure 1 and Appendix A.

During the treatment period, both male and female groups exhibited consistent changes in weight, with no statistically significant differences observed between the control and test substance groups (Figure 2a,b). During the recovery period, the 500 and 1000 mg/kg/day groups of females exhibited significantly lower body weights than the control group on days 7 and 14 of the recovery period, and the 1000 mg/kg/day group already showed significantly lower body weights than the control group on day 1 of recovery. The body weight gain during the recovery period was also significantly lower in both groups compared to the control group. In contrast, there were no changes in body weight and body weight gain among males.

### 3.3. Food Intake

The mean food intake and standard deviation for each group are shown in Figure 3. During the recovery period, the 1000 mg/kg/day male group exhibited a mild but statistically significant increase in food intake compared to the control group at 2 weeks of recovery (Figure 3a). In females, the 500 mg/kg/day group displayed a significant decrease in food intake at 1 week of recovery, and the 1000 mg/kg/day group showed a slight but statistically significant decrease at 1 and 2 weeks of recovery compared to the control group (Figure 3b).

### 3.4. Urinalysis

During the treatment period, 2 out of 10 females in the 1000 mg/kg/day group exhibited deep yellow urine coloration, which was found to be statistically significant (Appendix A). No other changes were observed. Additionally, deep yellow urine coloration was observed in 1 out of 5 males and females in the control group during the recovery period. This observation was considered to have no toxicological significance. There were no statistically significant differences in any of the recovery periods between the control and test substance groups.

### 3.5. Hematological Examination

Hematological examination results for each group are presented in Figure 4 and Appendix A. At the end of the treatment period, a significant increase in prothrombin time (PT) and activated partial thromboplastin time (APTT) was observed in the 50 mg/kg/day group of males; however, this was considered an incidental variation since it was not dose-related (Figure 4n,o). At the end of the recovery period, the number of eosinophils was significantly higher in the 1000 mg/kg/day group of males (Figure 4l). Although there was a significant decrease in PT in the 500 mg/kg/day group of females, it was also considered an incidental variation since this change was not of toxicological significance and was not dose-related (Figure 4n).

### 3.6. Blood Biochemical Tests

The results of blood chemistry tests for each group are shown in Figure 5 and Appendix A. At the end of the treatment period, no statistically significant differences were found between the control and test groups in any of the parameters (Figure 5a–t).

At the end of the recovery period, bile acid (TBA) and urea nitrogen (BUN) levels were significantly higher in the 1000 mg/kg/day group of males (Figure 5e,f). Additionally, significantly lower levels of calcium (Ca) were observed in the 1000 mg/kg/day group of females. These changes were considered minor and not attributable to the test substance.

### 3.7. Pathological Examination

#### 3.7.1. Necropsy

At the end of the treatment period, a unilateral cyst was observed in the kidney of one male in the control group. In females, discolored spots were observed in the stomach of the 500 mg/kg/day group, and bilateral dilated uterine horns were observed in one case in the 150 mg/kg/day group. (Appendix A)

At the end of the recovery period, one male in the 1000 mg/kg/day group had a diverticulum in the small intestine, one male in the 500 mg/kg/day group had discolored spots in the lungs, and one male in the control group had discolored spots in the stomach. In females, bilateral dilation of the uterine horns was observed in three cases in the 500 mg/kg/day group. (Appendix A)

#### 3.7.2. Organ Weights

The organ weights (absolute weights) for each group are shown in Figure 6 and Appendix A. At the end of the treatment period, a significantly higher brain weight was observed in the 500 mg/kg/day group of females (Figure 6a). However, since there was no clear dose–response relationship and the change was mild, it was considered an incidental variation.

At the end of the recovery period, significantly lower thyroid weights were observed in the 500 and 1000 mg/kg/day groups of males (Figure 6h). These changes were not dose-related and were considered incidental variations.

In females, significantly lower body weights were observed in the 500 and 1000 mg/kg/day groups, with significantly lower weights of the liver and pituitary gland (Figure 6c,g), as well as significantly higher relative brain weights, noted in the 1000 mg/kg/day group (Figure 6a).

### 3.8. Histopathological Examination

Histopathological examination of each group revealed changes in various tissues and organs at the end of the treatment period in both male and female groups treated with the test substance (Appendix A). However, the frequency of these changes was not statistically significantly different from that in the control group. Therefore, these changes were not attributed to the test substance treatment.

## 4. Discussion

Despite the needle-like crystal structure of fib-CNFs shown in Figure 1c, no damage to digestive organs was observed in the results of Section 3.6 and Section 3.7. Marcuello et al. [18] reported that the Young’s modulus of CNFs decreases in wet conditions. Therefore, it is necessary to measure the Young’s modulus of fib-CNFs in wet and acidic pH conditions using atomic force microscopy (AFM).

In discussing the safety of CNFs, Nadia et al. [19] reported that endotoxin (lipopolysaccharide, LPS) is a component of the cell wall of Gram-negative bacteria and is responsible for various pathological conditions in bacterial infections. Moriyama et al. [20] described the presence of endotoxin in CNFs. Menas et al. [21] used the Pierce LAL Chromogenic Endotoxin Quantitation Kit (Thermo Fisher Scientific, Waltham, MA, USA) to measure endotoxin contamination in five different nanocellulose materials, including lyophilized CNF powder. Endotoxin levels were reported to be below the detection limit (0.01 EU/mL).

Nordli et al. [22] developed a method to produce ultra-pure CNF with low endotoxin levels. For endotoxin testing, they used a modified QCL-1000 Endpoint Chromogenic LAL assay (Lonza, Allendale, NJ, USA) with a reduced amount of reagent. Their results showed that sodium hydroxide treatment and washing with Milli-Q water significantly reduced the LPS concentration from 8240 to 237 EU/g. Subsequent oxidation treatment via TEMPO further reduced the LPS concentration to 45 EU/g. Nordli et al. [22] reasoned that TEMPO-oxidized CNF underwent TEMPO-mediated oxidation under alkaline conditions during production, making the presence of LPS in the sample unlikely.

The mechanically fibrillated CNF used in this study was prepared using sodium hydroxide treatment and pure water, similar to Nordli et al.’s method. The CNF was also sterilized at 80 °C and subsequently sealed. The concentration of LPS in fib-CNF in this study was estimated to be 100 EU/g, suggesting a low endotoxin effect [23]. Therefore, the influence of LPS on the toxicity of fib-CNF ingestion in this study appears negligible.

The weight suppression effect of CNF administration can be attributed to two factors: the degradation of CNF by intestinal bacteria and the stimulation of the brain and pituitary gland by CNF.

First, the relationship between intestinal bacteria, the intestinal environment, and CNF is discussed. It is possible that the effect of fib-CNF on intestinal bacteria led to weight loss. Numerous studies and review articles discuss the degradation of cellulose and other dietary fibers by intestinal bacteria [24,25,26,27,28,29,30,31,32,33,34]. Although cellulose is a central component of plant fiber and is abundant in diets with plant-derived foods, humans, like other mammals, rely on gut microbiota to break down cellulose. Fermentation of dietary fiber by cellulosome-producing bacteria converts these indigestible compounds into short-chain fatty acids that contribute to host health, including the prevention of colon cancer and regulation of blood sugar levels.

Dietary fiber is also beneficial for maintaining the stability and diversity of the intestinal microbiota. However, the modern diet in the developed world is dominated by processed foods, is extremely low in plant fiber, and shows little evidence of cellulose breakdown and fermentation in the human intestinal tract.

The weight-inhibitory effect of CNF administration was examined by Nagano et al. [35] using mice. They suggested that the fibrous CNF used, similar to what we used, had an inhibitory effect on obesity mediated through the regulation of the intestinal microbiota equilibrium. The fibrous CNF they employed was BiNFi-s, WFo-10002, characterized by a standard fiber length of 2 μm, while the one used in this study had a shorter fiber length of less than 1 μm.

They orally administered 0.1 wt% and 0.2 wt% concentrations of the fibrous CNF solution in drinking water to male C57BL/6N mice for 7 weeks. Although this method of CNF administration through free water is straightforward, the amount of CNF ingested is estimated solely based on the volume of drinking water, making it difficult to quantify CNF intake due to spillage. The mice were also fed a high-fat diet (HFD). The interaction between the HFD and CNF is not yet fully understood.

In Nagano’s study [35], only mice fed with 0.2 wt% fibrous CNF exhibited a suppressive effect on body weight gain. He attributed this weight suppression effect to the regulation of intestinal microflora balance by fibrous CNF, which resulted in increased intestinal bacterial diversity, alterations in microflora composition, decreased relative abundance of *Streptococcaceae* and *Lakenaceae*, and increased relative abundance of *Lactobacillaceae*.

Zhang et al. [9] reported no significant differences in hematological and serum markers between the control group and mice fed with a CNF suspension. However, mice fed with the CNF suspension exhibited a significant increase in fecal fat content, along with decreased food intake and body weight, compared to the control group. These findings suggest that CNF is non-toxic and has potential as a food additive or supplement to reduce caloric intake.

Zhai et al. [36] investigated the effects of bacterial cellulose nanofibers (B-CNF) on diphenoxylate-induced constipation in rats. They observed a significant decrease in ATPase activity in the colons of constipated rats following B-CNF administration (*p* < 0.01). Histological examination of the colon showed that B-CNF administration effectively increased the length of villi cells and the thickness of the colonic mucosa and muscle (*p* < 0.01). Furthermore, B-CNF administration protected colonic smooth muscle cells from apoptosis. These findings suggest that B-CNF may serve as a promising dietary fiber to alleviate constipation. Therefore, CNF ingestion is expected to enhance the intestinal environment without causing toxicity.

In this study, changes in intestinal bacteria and the intestinal environment due to fib-CNF ingestion could not be detected through blood tests, necropsy observations, or weight changes. Therefore, additional experiments focusing on this aspect are necessary in the future.

Next, we will discuss the effects of fib-CNF on organs other than the brain and intestine.

Deviation from normal body weight gain in growing female rats was observed from the stage of fib-CNF administration, as shown in Figure 2b, but no significant difference was noted at this point. The suppression of body weight during the recovery phase was directly caused by a decrease in food intake (Figure 6g), which is assumed to be due to atrophy of the pituitary gland. Additionally, a reduction in liver weight was also noted (Figure 6c).

Blevins et al. [37] reported an association between the neurohypophyseal peptide oxytocin (OT) and weight control. The decrease in these organs may cause anorexia and weight loss, as reported by using Lindane [38]. Helmy et al. [39] showed in rats that chronic stress causes pituitary hormonal imbalance and weight loss. Li et al. [40] reported that intestinal stimulation lowered food intake and body weight in rats. The mechanism involved an appetite-promoting peptide in the hypothalamus. In the future, the relationship between fib-CNF intake and body weight will be further clarified; from our research, 500 mg/kg/day is considered to be the threshold level of fib-CNF at which dieting is possible without disturbing the pituitary gland or hormone balance.

Ide et al. [41] investigated the effects of dietary fiber on bile acids in rats. Non-soluble fibers, including cellulose, corn bran, and chitin (comprising 10% of the diet), and cholestyramine (5% of the diet) were found to have a slight inhibitory effect on bile acid secretion.

Nagano et al. [42] observed that mice fed with CNF displayed increased spontaneous locomotor activity. This suggests that the combination of CNF intake and increased physical activity may have a more substantial anti-obesity effect by reducing weight and fat gain, improving glucose tolerance, and balancing intestinal microbiota.

To examine whether fib-CNF affects the pituitary gland, individual pituitary weights of female rats were compared. Figure 7a shows absolute pituitary weights, and Figure 7b shows relative weights adjusted for body weight. In both cases, the pituitary gland weight increased with fib-CNF administration. Additionally, some individuals experienced significant pituitary gland weight loss when fib-CNF was discontinued. Further experiments with a larger sample size are necessary to determine whether the effects of fib-CNF on individual animals are significant.

It is unclear whether CNF is broken down by intestinal bacteria into β-1,4-linked glucose, which is then absorbed into the bloodstream and contributes to the loss of pituitary weight. Thus, further investigation is needed regarding the long-term oral administration of CNF in female rats.

## 5. Conclusions

Fib-CNF “BiNFi-s” FMa-10005, obtained through the mechanical defibrillation of pulp-derived cellulose, was administered via forced oral administration to male and female Crl:CD(SD) rats at doses of 50, 150, 500, and 1000 mg/kg/day for 28 days. The control group received water for injection. The study aimed to assess the toxic effects of repeated administration, and the rats were also monitored for 14 days post-administration to evaluate recovery from any toxic effects.

The results indicated that no deaths occurred in either sex during the administration period, and no toxicological effects related to the test substance were observed. Assessments included general condition and behavioral observations, urinalysis, hematological examination, blood biochemical examination, necropsy findings, organ weights, and histopathological examination.

Only female rats treated with 1000 mg/kg/day of CNF exhibited a consistently slow increase in body weight during the 14-day recovery period post-treatment. They also showed a slight decrease in pituitary and liver weights. However, hematological and blood biochemical tests did not reveal significant differences, suggesting a potential weight-suppressive effect of CNF ingestion. Although the health safety of oral CNF ingestion was confirmed, the cause of the slow increase in body weight during the recovery period in female rats needs to be elucidated in future studies.

## Figures and Tables

**Figure 1 nanomaterials-14-01082-f001:**
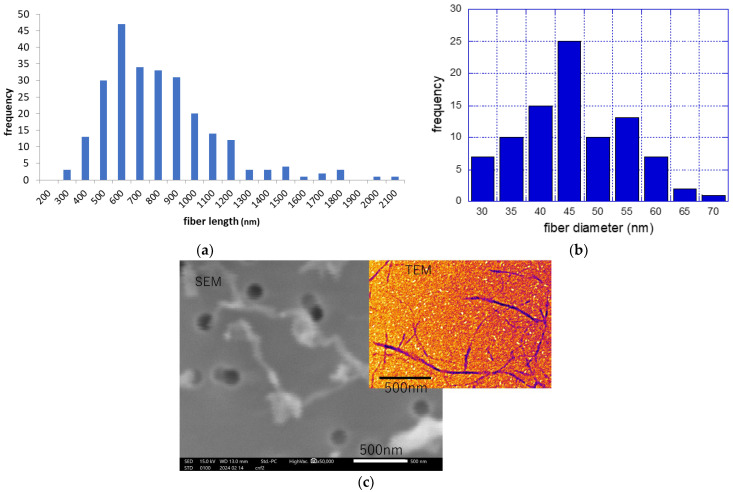
Characterization of BiNFi-s ultra-short FMa1005 fib-CNF. (**a**) Fiber length distribution. (**b**) Fiber diameter distribution. (**c**) SEM and TEM images showing the morphology of the fibers. Bars = 500 nm.

**Figure 2 nanomaterials-14-01082-f002:**
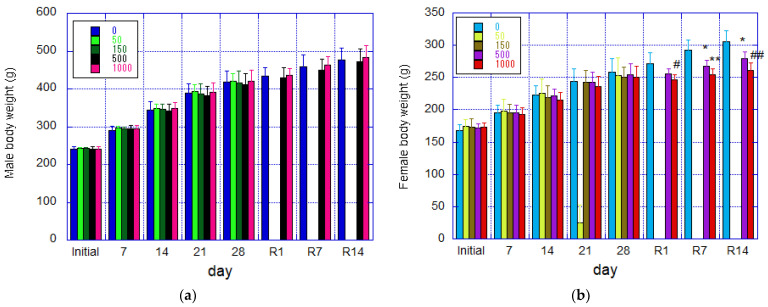
Mean body weights of rats during the 28-day administration and 14-day recovery periods (mean ± S.D.). (**a**) Male rats. (**b**) Female rats. Significant differences from the control group are indicated by * (*p* ≤ 0.05) and ** (*p* ≤ 0.01) using *t*-test; # (*p* ≤ 0.05) and ## (*p* ≤ 0.01) using Welch test. R denotes the recovery period.

**Figure 3 nanomaterials-14-01082-f003:**
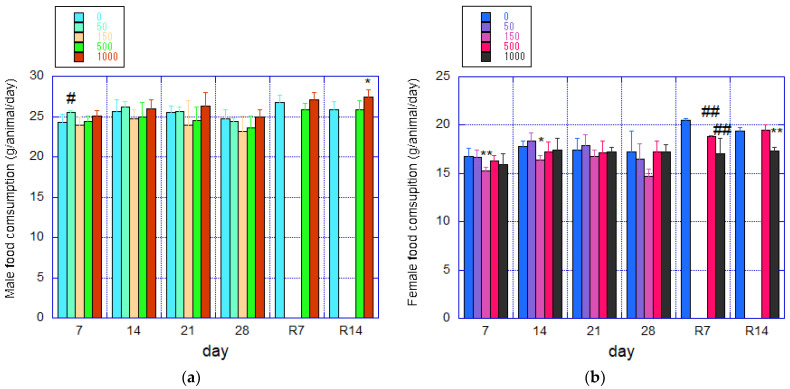
Mean daily food consumption (g/animal/day) during the 28-day administration and 14-day recovery periods (mean ± S.D.). (**a**) Male rats. (**b**) Female rats. Significant differences from the control group are indicated by * (*p* ≤ 0.05) and ** (*p* ≤ 0.01) using *t*-test; # (*p* ≤ 0.05) and ## (*p* ≤ 0.01) using Welch test. R denotes the recovery period.

**Figure 4 nanomaterials-14-01082-f004:**
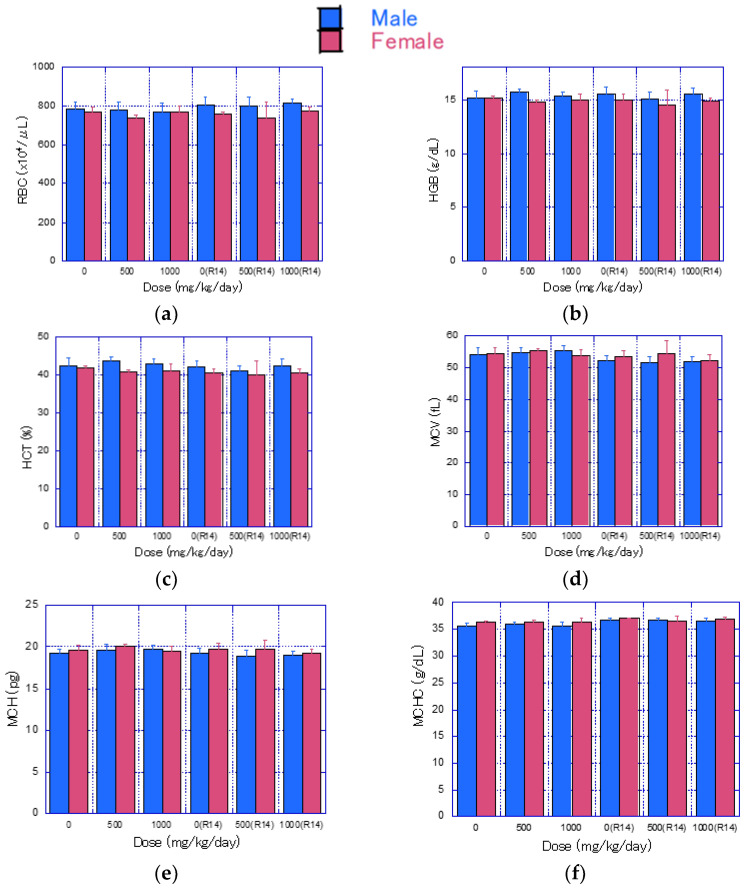
Mean values of hematological test parameters (mean ± S.D.) after 4 weeks of CNF administration and 2 weeks post-administration. Blue bars represent male rats, and red bars represent female rats. Significant differences from the control group are indicated by * (*p* ≤ 0.05) and ** (*p* ≤ 0.01) using *t*-test. (**a**) RBC, (**b**) HGB, (**c**) HCT, (**d**) MCV, (**e**) MCH, (**f**) MCHC, (**g**) PLT, (**h**) Reticulocyte, (**i**) WBC, (**j**) Lymphocyte, (**k**) Neutrophil, (**l**) Eosinophil, (**m**) Monocyte, (**n**) PT, (**o**) APTT.

**Figure 5 nanomaterials-14-01082-f005:**
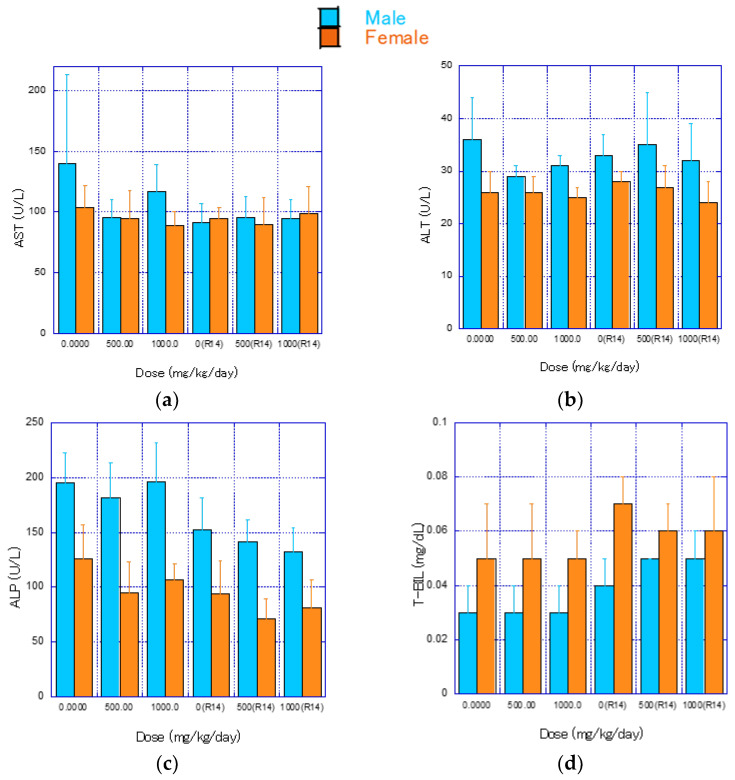
Mean values of blood biochemical parameters (mean ± S.D.) after 4 weeks of CNF administration and 2 weeks post-administration. Light blue bars represent male rats, and bitter orange bars represent female rats. Significant differences from the control group are indicated by * (*p* ≤ 0.05) using *t*-test. (**a**) AST, (**b**) ALT, (**c**) ALP, (**d**) T-BIL, (**e**) TBA, (**f**) BUN, (**g**) CRE, (**h**) GLU, (**i**) T-CHO, (**j**) PL, (**k**) TG, (**l**) TP, (**m**) ALB, (**n**) A/G ratio, (**o**) IP, (**p**) Ca, (**q**) Mg, (**r**) Na, (**s**) K, (**t**) Cl.

**Figure 6 nanomaterials-14-01082-f006:**
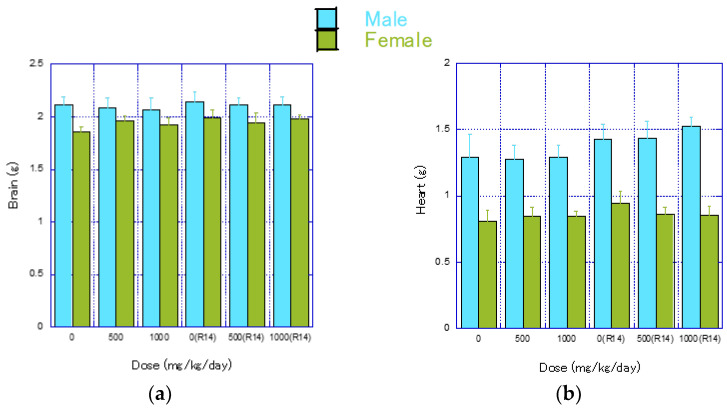
Mean values of organ weights (mean ± S.D.) after 4 weeks of CNF administration and 2 weeks post-administration. Blue bars represent male rats, and green bars represent female rats. Significant differences from the control group are indicated by * (*p* ≤ 0.05) using *t*-test. (**a**) Brain, (**b**) Heart, (**c**) Liver, (**d**) Kidneys, (**e**) Spleen, (**f**) Thymus, (**g**) Pituitary, (**h**) Thyroids, (**i**) Adrenals, (**j**) Testes, (**k**) Prostate, (**l**) Epididymides, (**m**) Seminal vesicles, (**n**) Ovaries, (**o**) Uterus.

**Figure 7 nanomaterials-14-01082-f007:**
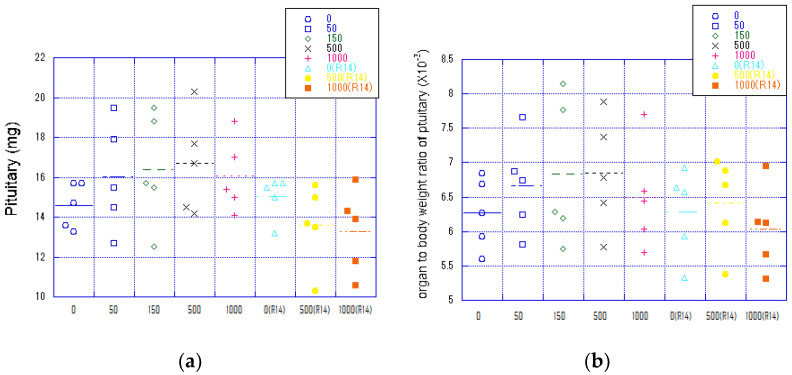
Changes in pituitary weight in female rats during fib-CNF administration and recovery. Dotted lines in figures indicate average of each value. (**a**) Absolute pituitary weights. (**b**) Weight ratio of the pituitary gland to the body weight of individual female rats.

**Table 1 nanomaterials-14-01082-t001:** Group composition and number of animals used.

Sex	Group	Dose mg/kg/day	Dosing Volume mL/kg	Dose Concentration wt%	Number of Animals
Male	Control	0 ^a^	10 mL, two divided doses	0	5
	Control recovery	0 ^a^		0	
	Low dose	50		0.25, two doses	
	Middle dose	150		0.75, two doses	
	High dose	500		2.5, two doses	
	High dose recovery	500		2.5, two doses	
	Dosage limit	1000		5.0, two doses	
	Dosage limit recovery	1000		5.0, two doses	
Female	Control	0 ^a^	10 mL, two divided doses	0	5
	Control recovery	0 ^a^		0	
	Low dose	50		0.25, two doses	
	Middle dose	150		0.75, two doses	
	High dose	500		2.5, two doses	
	High dose recovery	500		2.5, two doses	
	Dosage limit	1000		5.0, two doses	
	Dosage limit recovery	1000		5.0, two doses	

^a^: Dosing water medium.

**Table 2 nanomaterials-14-01082-t002:** Test Parameters and Methods for Urinalysis.

Test Parameters	Measuring Methods and Equipment
(1) pH	CLINITEK Advantus Urine Chemistry Analyzer Multistix^®^ 10 SG (Siemens Healthineers, Erlangen, Germany) Siemens Medical Solutions
(2) Blood	
(3) Ketone	
(4) Glucose	
(5) Protein	
(6) Urobilinogen	
(7) Bilirubin	
(8) Urine sediment	New Sternheimer stain microscope examination (Olympus BX46, Tokyo, Japan)
(9) Color tone	Observation with the naked eye
(10) Amount of urine	Weighing method Electronic balance LA4200 (Sartorius Co., Ltd., Göttingen, Germany)
(11) Specific gravity	ATAGO Serum Protein Flexure Meter N (ATAGO Co., Ltd., Tokyo, Japan)

**Table 3 nanomaterials-14-01082-t003:** Hematological Test Parameters and Measurements.

Test Parameters	Measurement Method
(1) Red Blood Cell Count (RBC)	Sheath flow DC detection method
(2) Hematocrit value (HCT)	
(3) Platelet count (PLT)	
(4) Hemoglobin (HGB)	SLS hemoglobin method
(5) Mean corpuscular volume (MCV)	Sheath flow DC detection and calculation method
(6) Mean corpuscular hemoglobin (MCH)	
(7) Mean corpuscular hemoglobin concentration (MCHC)	
(8) Reticulocyte count	Flow cytometry method and calculation method
(9) White Blood Cell Count (WBC)	Flow cytometry method
(10) White Blood Cell Count by Type Lymphocyte Count Neutrophil Count Eosinophil Count Basophil Count Monocyte Count	
(11) Prothrombin time (PT)	Solidification Time Method: Light Scattering Inspection Method
(12) Activated partial thromboplastin time (APTT)	

**Table 4 nanomaterials-14-01082-t004:** Items and methods of blood biochemical tests.

Test Parameters	Measurement Method
(1) Aspartate aminotransferase (AST)	JSCC recommendation method
(2) Alanine aminotransferase (ALT)	JSCC recommendation method
(3) Alkaline phosphatase (ALP)	IFCC Reference Measurement Operation Method
(4) Total pililpine (T-BIL)	Vanadate oxidation
(5) Bile acid (TBA)	Enzymatic cycling
(6) Urea nitrogen (BUN)	UV method (urease/GLDH method)
(7) Creatinine (CRE)	Enzymatic process
(8) Glucose (GLU)	HK-G-6-PDH method
(9) Total cholesterol (T-CHO)	Cholesterol oxidase method
(10) Phospholipids (PL)	Choline oxidase/DAOS method
(11) Triglycerides (TG)	FG scavenging enzyme method
(12) Total protein (TP)	Pioulet method
(13) Albumin (ALB)	BCG method
(14) A/G ratio (albumin/globulin ratio)	Computation method
(15) Inorganic phosphorus (IP)	Enzymatic process
(16) Calcium (Ca)	Enzymatic process
(17) Magnesium (Mg)	Xylidiliriproof method
(18) Sodium (Na)	Ion selective electrode method
(19) Potassium (K)	Ion selective electrode method
(20) Chlorine (Cl)	Ion selective electrode method

## Data Availability

Data are contained within the article and Appendix A.

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
