# Peer review of "A 28-Day Repeated Oral Administration Study of Mechanically Fibrillated Cellulose Nanofibers According to OECD TG407"

_nanomaterials, 2024, doi:10.3390/nano14131082_

Round 1
Reviewer 1 Report
Comments and Suggestions for Authors
This paper presents a significant evaluation of the toxicity of ingesting fibrillated nanocellulose (fib-CNF). The authors meticulously conducted experiments using rats over a four-week period, yielding a diverse range of results. The comprehensive nature of the paper, with its extensive results and statistical analysis, underscores the importance of this research in our understanding of nanocellulose toxicity.
It is exciting to note that CNF did not cause any toxicity in rats, only weight suppression.
1. One crucial aspect that needs to be addressed is the purity of the CNF ingested by the rats. It's essential to know if the CNF was free of impurities, such as heavy metals from the process or nitrogen, sulfur, etc., that could potentially impact the results. Data on the elemental content, mineral content, ash content, etc. of the CNF would be highly beneficial, especially since some weight loss effects were observed.
2. The paper needs further revision in terms of formatting. In Figures 4-6, many results are graphed, but there is no explanation of the color of the graph columns. It is only described in the figure captions. Since the graphs are spread over two to three pages, a legend for the column colors should be shown on each graph. Also, the serial numbers (alphabets) assigned to each graph must be closer. This can lead to a misunderstanding that each alphabet refers to the graph under that alphabet. Please correct the labeling to avoid confusion.
Author Response
Thank you very much for your kind review. We have made the following changes to the manuscript and appreciate your review.
1) The fib-CNF used in this study contained less than 0.01 mg of elements, minerals, and ash per kg of fib-CNF solids after quantification of metal content by ICP-MS. These were added.
2)Explanation of the graph columns have been added to the top of the first graph in Figures 4-6. The alphabet is positioned in the upper left corner of the figure based on the paper's template. One line of space was added at the bottom of the figure.
Reviewer 2 Report
Comments and Suggestions for Authors
The manuscript titled “28-day repeated oral administration study of mechanically fibrillated cellulose nanofibers (fib-CNF) according to OECD TG407” by Yamashita, Y.; et al. is a scientific work where the authors monitored the toxicity effect related to the administration of cellulose nanofibers in rat models. Many clinical tests were carried out to shed light about the impact of fib-CNF on relevant metabolite levels. The most reventant finding obtained in this research could be interesting for a specialized target audience. Furthermore, the manuscript is generally well-written.
However, it exists some aspects that need to be addressed (please, see them below detailed point-by-point). The most relevant outcomes remarked by the authors can contribute in the growth of many fields by the precise design of the next-generation of therapies against liver diseases. For this reason, I will recommend the present scientific manuscript for further publication in Nanomaterials once all the below described suggestions will be properly fixed.
Here, there exists some points that must be covered in order to improve the scientific quality of the manuscript paper:
1) KEYWORDS. The authors should consider to modify the term “TG407” by “OECD TG407” in the keyword list.
2) INTRODUCTION. This section clearly depicts the current state-of-the-art. No actions are requested from the authors.
3) EXPERIMENTS. “The fiber length (…) analysis software” (lines 110-113). What was the settled acceleration electron voltage to gather the SEM images?
4) RESULTS. “3.1. Fiber Length and Fiber Diameter of Fib-CNF” (lines 259-269). Figure 1, panels a, b (line 265). The gaussian fitting with the respective regression coefficient (R2) should be furnished.
5) Then, in this subsection it may be opportune to remark the high-moisture stability of individual cellulose fibers [1] and how this aspect favours the design of innovative cellulose nanostructures for the smart delivery of selected compounds [2].
[1] Marcuello, C.; Foulon, L.; Chabbert, B.; Aguié-Béghin, V.; Molinari, M. Atomic force microscopy reveals how relative humidity impacts the Young’s modulus of lignocellulosic polymers and their adhesion with cellulose nanocrystals at the nanoscale. Int. J. Biol. Macromol. 2020, 147, 1064-1075. https://doi.org/10.1016/j.ijbiomac.2019.10.074.
[2] Liu, S.; Qamar, S.A.; Qamar, M.; Basharat, K.; Bilal, M. Engineered nanocellulose-based hydrogels for smart drug delivery applications. Int. J. Biol. Macromol. 2021, 30, 275-290. https://doi.org/10.1016/j.ijbiomac.2021.03.147.
6) “3.2. Weigth-3.6.2. Organ weights (lines 271-357). Why did the authors only test the first 14 days after the fibrillated cellulose nanofiber administration and not longer scale times? A brief statement should be furnished in this regard.
7) DISCUSSION. This section perfectly remarks the most relevant outcomes found by the authors in this work. No actions are requested from the authors
8) CONCLUSION. Here, the authors should add a brief statement to discuss about the future line actions to pursue this research and the open perspectives.
Comments on the Quality of English LanguageThe manuscript is generally well-written albeit it may be desirable if the authors could do a final check to polish those details susceptible to be improved.
Author Response
Thank you very much for your kind review. We have made the following changes to the manuscript and appreciate your review.
1)Corrected to OECD TG407.
2) The introduction section was reviewed and revised to include only objective background.
3) The acceleration voltage of the measured electron microscope is added.
4) The variation of fiber length and fiber diameter is described by R2 in Figure 1.
5)Two references [1] were cited in the Discussion. [2] were cited in the introduction.
6)I have added the following explanations: TG407 recommends 28 days of oral administration followed by 14 days of observation for recovery; TG408 is a 90-day oral administration study. I added that TG408 examination is also necessary for practical use in the future.
That is, TG407 study is a 28-day continuous oral administration. It is decided to use a minimum of 10 animals (5 females and 5 males each) for each dose. The addition of a satellite group of 10 animals each (5 males and 5 females) to the control and highest dose groups is to be considered in order to observe the reversibility, persistence or delayed onset of toxic effects for at least 14 days after completion of administration. Therefore, the recovery period was set at 14 days. TG408 is a continuous dosing study for more than 90 days, and in this case, there is no recovery period.
7)In the discussion section, we focused on the results obtained in this study and deleted irrelevant parts.
8)Conclusions were also added regarding the need for future research.
9)Native checking of the English was performed.